# Contrasting Pollination Strategies and Breeding Systems in Two Native Useful Cacti from Southern Brazil

**DOI:** 10.3390/plants12061298

**Published:** 2023-03-13

**Authors:** Rafael Becker, Oscar Perdomo Báez, Rosana Farias Singer, Rodrigo Bustos Singer

**Affiliations:** 1Laboratory of Systematics of Vascular Plants, Postgraduate Program in Botany, Universidade Federal do Rio Grande do Sul, Porto Alegre 91509-900, RS, Brazil; 2Porto Alegre Botanical Garden, Secretaria do Meio Ambiente e Infraestrutura do Estado do Rio Grande do Sul, Porto Alegre 90119-900, RS, Brazil

**Keywords:** Apidae, Cactaceae, *Cereus hildmannianus*, Coleoptera, conservation, *Pereskia aculeata*, plant reproduction, Sphingidae

## Abstract

Brazil is one of the centers of diversity of Cactaceae, yet studies addressing both pollination biology and the breeding system in Brazilian cacti are scarce. We herein present a detailed analysis of two native species with economic relevance: *Cereus hildmannianus* and *Pereskia aculeata*. The first species produce edible, sweet, spineless fruits and the second species produces leaves with high protein content. Pollination studies were undertaken through fieldwork observations in three localities of Rio Grande do Sul, Brazil, over two flowering seasons, totaling over 130 observation hours. Breeding systems were elucidated utilizing controlled pollinations. *Cereus hildmannianus* is solely pollinated by nectar-gathering species of Sphingidae hawk moths. In contrast, the flowers of *P. aculeata* are pollinated by predominantly native Hymenoptera but also by Coleoptera and Diptera, which gather pollen and/or nectar. Both cacti species are pollinator-dependent; neither intact nor emasculated flowers turn into fruit, yet whereas *C. hildmannianus* is self-incompatible, *P. aculeata* is fully self-compatible. In sum, *C. hildmannianus* is more restrictive and specialized regarding its pollination and breeding system, whereas *P. aculeata* is more generalist. Understanding the pollination needs of these species is a necessary starting point towards their conservation but also for their proper management and eventual domestication.

## 1. Introduction

Cactaceae is one of the most easily recognizable plant families, embracing about 2230 species distributed in about 176 genera [1]. Except for *Rhipsalis baccifera* (Sol.) Stearn, which occurs naturally in Africa and Sri Lanka, all remaining cacti are restricted to the Americas [1]. Morphological and physiological adaptations such as the presence of CAM metabolism and reduced axillary meristems to areoles and leaves reduced to spines, allowing this family to thrive under xeric to desertic conditions [1]. Nevertheless, cacti can also be found in tropical forests and rocky outcrops in grasslands, and this reflects in the diversity of forms, sizes, and floral morphology [1]. Most cactus diversity is found in Mesoamerica where native communities frequently benefit from products (normally fruits but also edible stems) of wild, semi-domesticated, or domesticated species as well [2]. Brazil is one of the centers of cacti diversity, with 37 cacti genera and 227 species [1,3]. It is the Brazilian Northeastern region that is concentrated with the most species diversity and, consequently, most ethnobotanical uses of Brazilian cacti [4]. In contrast, the southern Rio Grande do Sul State has 11 genera and 65 species (ca. of 29% of the Brazilian species) [3]. This Brazilian state is remarkable because it holds two contrasting biomes: the Atlantic Rain Forest and the Pampas. The Pampas Biome, in Brazil, is restricted to Rio Grande do Sul State but is also shared with neighboring countries such as Argentina and Uruguay [3,5].

The Pampa vegetation has always been neglected until recent research revealed remarkable records of plant diversity [5]. Of 65 cacti species native to Rio Grande do Sul, 53 (81.53%) are threatened with extinction according to IUCN criteria and most of them occur in rocky outcrops of the Pampa [3]. The main reasons for these numbers are twofold: habitat loss and illegal collection by poachers [3]. Understanding both the reproductive system and the dependence (or lack of) on pollinators is essential not only for the conservation of plant species in situ but also for the proper management of plant species of economic value [6]. Only a few studies have addressed pollination in Cactaceae from Southern Brazil [7,8,9] and only one has coupled pollinator observations and breeding system experiments [7]. However, even though the studies did not identify the pollinators, they pointed out the relevance of native solitary bees for the pollination of these cactus species [8,9].

The present contribution aims to present a detailed study of the breeding system and natural pollination of two cactus species native to Rio Grande do Sul: (1) *Cereus hildmannianus* K. Schum. and *Pereskia aculeata* Mill. *Cereus hildmannianus* is popularly known as “Tuna” and its edible, sweet, spineless fruits make them valuable for horticultural purposes [10]. A very similar species, *Cereus peruvianus* (Pfeiff.) K. Schum., has recently been established under cultivation for such purposes in Israel, where it is sold under the commercial name of “Koubo” [11]. Furthermore, several ethnomedicinal uses (against pulmonary disorders, rheumatism, etc.) of extracts taken from the stems were identified, as well as several promising pharmacological compounds [12]. (2) *Pereskia aculeata* is popularly known as “ora-pro-nobis” (In Latin, it means “pray for us” because the plant normally flowers during or close to Easter). Unlike most Cactaceae, this species presents well-developed, ovate leaves. In turn, these leaves present a high protein content and are highly appreciated in culinary usage [13,14,15,16]. The species also shows medicinal properties, presenting compounds that either foster the immunological system, antibacterial activity, or antiproliferative effect in neuroblastoma cells [17,18].

The questions behind this contribution are the following: (1) Are the study species pollinator-dependent? (2) If so, which are the pollinators? (3) Which is the breeding system in the study species? (4) How do pollinators contribute to cross-pollination?

Our preliminary observations suggested that the natural fruit set is rare in *C. hildmannianus*. According to the preceding bibliography [3], the flowers open at night, and floral morphology strongly suggests pollination by long-tongued Sphingidae hawk moths, as is already documented for other *Cereus* species [19,20]. Conversely, preliminary observations of the fruit set of *P. aculeata* at the Porto Alegre Botanical Garden indicated that fruits are abundantly produced. According to Carneiro et al. (2016), the flowers are diurnal and strongly scented. Their color and shape strongly suggest bee pollination. In this context, we established the following hypotheses: (1) Both study species may be pollinator-dependent (this is, may need the agency of pollinators to set fruit and viable seed). (2) Owing to the apparent rarity of fruits, *C. hildmannianus* may be self-incompatible (unable to set fruit when pollinated with pollen of the same individual). (3) Owing to the commonness of fruits in *P. aculeata*, this species may be self-compatible. (4) Owing to its flower features and according to preceding literature in related species, *C. hildmannianus* may be pollinated by Sphingidae hawk moths. (5) Owing to its flower features, *P. aculeata* may be pollinated by bees.

## 2. Results

### 2.1. Overall Flower Features and Nectar Concentration

The flowers of *Cereus hildmannianus* have a nocturnal anthesis, with the flowers starting to open around about 9:00 PM and fading between 6:00 AM and 9:00 AM, depending on climate conditions. The studied flowers showed a color variation on the external side of the tepals, varying from white to soft pink (Figure 1a–c). Sometimes this variation was observed in the same individual. In studied plants, the length of the flower varied between 16 and 20 cm (mean = 18.2 cm ± 1.4) (Figure 1c). The nectar does not form a column at the bottom of the floral tube. Instead, it is found as dense droplets covering the hypanthium (Figure 1c). The nectar volume found was between 88 and 104 μL (mean = 94.8 μL ± 2.38) and the sugar concentration varied from 15 to 29% (mean = 23.4% ± 1.67) (*n* = 10 individuals; 50 flowers). The flowers of *P. aculeata* present diurnal anthesis, opening around 6:00 AM and closing around 1:00 PM. Perianth’s color is normally white (Figure 1d–f), exceptionally light cream, and the filaments of the stamens can be yellow or red (Figure 1d–f). Flowers do not form a tube and the perianth width measured was 5.2 to 5.8 cm (mean = 5.5 ± 0.22). The nectar is found just below the stamens. Nectar volume was lower than 0.01 mL. The average sugar concentration varied between 6 and 15% (mean = 10.6% ± 4.77) (*n* = 4 individuals; 20 flowers).

### 2.2. Breeding Systems

Both studied species are clearly pollinator-dependent since no fruit was obtained from either intact or emasculated flowers (Table 1). However, all self-pollinated flowers of *C. hildmannianus* aborted, indicating that this species is self-incompatible. Conversely, *P. aculeata* produced a high fruit set from either self-pollinated or cross-pollinated flowers and no statistically significant differences were observed between these treatments (*p*-value = 0.5958). Thus, *P. aculeata* can safely be considered fully self-compatible. The natural pollination success for both species was high: 70% in *C. hildmannianus* in São Francisco de Assis population (21/30 fruits produced) and 100% for *Pereskia aculeata* (Table 1).

### 2.3. Pollinators, Pollinator Behavior, and Pollinator Frequency

At the flowers of *Cereus hildmannianus*, we observed three species of nectar-seeking hawk moths (Sphingidae) as pollinators: *Eumorpha satellitia* (Linnaeus 1771) (Figure 2e,f), *Eumorpha vitis* Linnaeus 1758 (Figure 2c), and *Manduca sexta* (Linnaeus 1763) (Sphingidae, Lepidoptera) (Figure 2b,d) (Appendix A). At night, Cetoniinae beetles were observed copulating inside the floral tube but without touching the stigma. Thus, these beetles can only be considered floral visitors. At dawn, honeybees, bumblebees (*Bombus pauloensis* (Friese 1912)), and stingless bees (*Trigona spinipes* (Fabricius 1793)) were observed collecting pollen but in no instance did these bees touched the stigma and their passive behavior suggests that they are solely non-pollinating flower visitors. As a whole, we counted 44 hawk moth visits during the whole observation period. The photographic and film record indicates that all moth species uncoil their proboscises before landing into the flowers and that they use their forelegs to guide themselves into the floral tube (Figure 2a,b). Inside the flower, the hawk moths contact the stamens and the stigma with the ventral region of their body and wings (Figure 2c–f) (Appendix A). The hawk moths spent up to between 15 and 260 s (mean = 47 s ± 14.57) at the flowers. The observed peak of visiting happened between midnight and 1:00 AM. Before 11:00 PM, the hawk moths hovered around the flowers without entering the floral tube. As for the pollinator frequency, during observations in São Francisco de Assis, hawk moths visited the flowers regularly, at 35–50 min intervals, between 11:00 PM and 2:00 AM. (Figure 3).

Anthesis in *P. aculeata* starts at 6:00 AM and lasts until 1:00 PM, but pollinator visits became rare after midday. During the observation period, we recorded insects of the orders Hymenoptera, Coleoptera, and Diptera as pollinators. The Hymenoptera were far more abundant (Table 2) and were represented by bumblebees (*Bombus morio* (Swederus 1787)—Figure 4a—and *B.* pauloensis—Figure 4b), carpenter bees (*Xylocopa frontalis* (Olivier 1789)—Figure 4c), introduced honeybees, small-sized stingless bees (*Plebeia droryana* (Friese 1900)—Figure 4e, *Scaptotrigona bipunctata* (Lepeletier 1836)—Figure 4d, and *T. spinipes*), and wasps of *Polybia* sp. (Vespidae—Figure 4g). The Coleoptera were represented by beetles of *Caryedes* sp. (Chrysomelidae: Bruchini) (Figure 4f) and Diptera by unidentified Syrphidae flies. Owing to our photographic and film record, all these insects were effective pollinators; that is, contacting both androecium and gynoecium during their visits. Even insects disproportionally small in comparison with the flowers, such as workers of *P. droryana,* were recorded hovering around the flowers while transferring the pollen and landing on the stigmatic surface (Appendix A). Based on the photographic and film record, all bee species collected both pollen and nectar (Table 2 and Appendix A), wasps foraged solely on nectar, and *Caryedes* sp. and Syrphid flies foraged only on pollen (Table 2). Nectar-seeking insects routinely inserted their proboscises into the floral disc, below the stamens. While gathering pollen, bees mixed it with saliva and stored it in their corbiculae (Apidae) or scopae (Halictidae), where it was very visible as orange packs (Figure 4a,b,d,e). Two species of Lepidoptera (*Urbanus* sp. (Hesperiidae) and *Aellopos* sp. (Shingidae)) were recorded only once gathering nectar at the flowers. During these brief visits, it was not possible to establish if they were, in fact, pollinators, but their infrequency indicates that is unlikely that they are important for the fruit set of *P. aculeata*; at least at the study site.

When recording pollinator frequency, a total of 246 visits were documented. About 66% of these interactions were made by stingless bees (Meliponini) (Table 2). All other insects were less frequent. The exotic *Apis mellifera* (Apini) visited the flowers 33 times, which was more than bumblebees and carpenter bees but considerably less than native stingless bees (Table 2). The frequency of interaction for each species by one-hour intervals is described in Figure 5. The peak of visiting was observed between 9:00 and 10:00 AM, with 45 interactions. Intervals with more pollinator diversity were between 9:00 and 10:00 AM and between 10:00 and 11:00 AM, with seven species. Native bees were the top interactive group in all intervals, with *T. spinipes* and *S. bipunctata* leading the interaction between 7:00 and 12:00 AM. Some species, such as *Polybia* sp. and *Augochlora* sp. (Appendix A) were observed during the first and last hours of anthesis, respectively. The average time of permanency at the flowers varied between species, without relation to frequency (Table 2). *Caryedes* sp. was the species with the most durable interaction (up to 104 s), while *B. pauloensis* and *B. morio* spent about 3–4 s per visit. The native bees *T. spinipes* and *S. bipunctata* spent a mean of 35 and 32 s, respectively. It is an intermediate average time compared to the other bee species which spent less than 10 s or more than 50 s.

## 3. Discussion

This is the first study coupling detailed pollination observations and breeding system experiments in *C. hildmannianus* and *P. aculeata*. To the best of our knowledge, this is also the first pollination report for the whole cactus subfamily Pereskioideae, as it is currently defined [21]. Only a handful of preceding studies in Cactaceae from Southern Brazil have been published to date [7,8,9] and only the work of Cerceau et al. (2019) presented both a detailed account of pollination behavior and breeding systems. The other two preceding studies [8,9] focused on pollinator behavior alone.

### 3.1. Overall Flower Features

As it is a rule in Cactaceae, both studied species presented ephemeral flowers that last less than one day. However, the two studied species present contrasting flower shapes (tubular vs. open, disk-like) and phenologies (nocturnal vs. diurnal flowering). This is reflected in completely different sets of pollinators. As expected, the open-disk-like flowers of *P. aculeata* received a more diverse array of pollinators than the morphologically more complex flower of *C. hildmannianus.* An early divergent group, such as *Pereskia*, tends to show generalist morphology [22]. Therefore, more specialized pollination strategies (such as ornithophily or sphingophily) are to be expected in more internal and more recently-diverged groups [23]. The flowers of *C. hildmannianus* present a set of features (nocturnal anthesis, long, fragrant, nectariferous flowers, with the nectar placed inside this tube) that fit well in the syndrome of sphingophily (pollination by hawk moths) [24], a fact that was confirmed by this work. This floral ground plan allows only a specific group of visitors (in this case, Sphingidae) to access the nectar reward with their long proboscises. In addition, nectar concentration (ca. 23%) is similar to those found for *C. peruvianus* (ca. 27%) [19], *Cereus fernambucensis* Lem. (ca. 24%) [20], and other species with Sphingidae pollinators. The floral ground plan in *P. aculeata* makes their flowers less restrictive. Overall flower features (diurnal, fragrant, disk-like, brightly-colored flowers) likely fit well under the syndrome of mellitophily (bee pollination). However, insects other than bees were also observed as effective pollinators (see Results). The nectar concentration (ca. 10%) was lower than expected for “typical” melitophylous flowers (ca. 30%) [24]. Indeed, it is important to remember that floral syndromes, as defined by Faegri and van der Pijl (1979), are hypotheses that have to be tested through fieldwork. There are well-documented cases in Cactaceae [25] where floral syndromes and actual pollinators do not match. The flowers of the Andean *Denmoza rhodacantha* (Salm-Dyck) Britton & Rose (Cereeae: Trichocereinae) present features suggesting bird pollination, yet the actual pollinators are Halictidae bees [25]. A similar mismatch has been documented on the pollination of the columnar *Trichocereus pasacana* (F.A.C. Weber) Friederich & Rowley, whose flowers are nocturnal and apparently sphingophilous but are apparently pollinated by carpenter bees [26]. Indeed, there are several reports of cacti admitting more than one class of pollinators. This is the case of *Cipocereus laniflorus* N.P. Taylor & Zappi [27], of *Echinopsis chiloensis* subsp. *chiloensis* (Colla) Friederich & Rowley [28], of *Echinopsis ancistrophora* Speg. [29], and *Echinopsis terscheckii* (Parm. ex Pfeiff.) Friederich & Rowley [30]. The first species is mainly pollinated by bats (but also by insects) [27] and the three *Echinopsis* species are mainly pollinated by Sphingidae moths but also by other insects (especially bees) [27,28,30]. It has been suggested that floral syndromes rather than indicating all the possible pollinators do indicate the more effective ones [31]. The same authors have suggested that secondary pollinators often represent the ancestral pollination condition [31]. Experiments promoting the selective exclusion of a class of pollinators (e.g., excluding big-sized bees) could be used to test these ideas in cacti with generalistic pollination, such as *P. aculeata*.

### 3.2. Breeding Systems

Both studied species are pollinator-dependent. Neither intact nor emasculated flowers set fruit, indicating that flowers do not undergo automatic self-pollination or develop fruit from apomixis. This points towards the necessity of pollinators’ agency for the plants to set fruit and viable seed. We herein report that *C. hildmannianus* is self-incompatible; this is, unable to set fruit following pollination with the pollen of the same individual. Self-incompatibility has already been observed in *C. peruvianus* [19] and *C. repandus* [32]. Conversely, *C. fernambucensis* with 45% of fruiting rate in manual self-pollination treatment [20] is considered partially self-compatible. It has been suggested that in Cactaceae that there is a correlation between self-compatibility and life forms [31]. Tree-like, shrub-like, and columnar species (as *C. hildmannianus*) usually have a long-life cycle and would tend towards self-incompatibility. In contrast, globose species that usually have short life cycles and produce few flowers need to be self-compatible to guarantee reproductive success [31]. If this pattern proves true, it is to be expected that self-incompatibility be prevailing within the tree-like, columnar cacti of Cereeae (*Cereus* and related genera). In contrast, *P. aculeata* has proven to be self-compatible and we did not detect statistical differences between the percentages of fruits developed either through manual self-pollination or cross-pollination. *Pereskia guamacho* F.A.C. Weber, a species from the Caribbean region has proven to be self-incompatible [33]. However, it is important to stress that this is one of the species that, according to phylogenetic analyses, are to be placed in another subfamily, namely, Leuenbergioideae (as *Leuenbergeria guamacho* (F.A.C. Weber) Lodé), and, consequently, is not a “true” *Pereskia*. Unfortunately, the literature regarding the breeding system of *Pereskia sensu stricto* is scant and more species should be studied to obtain a more informative background. In this context, this contribution is the first report of the breeding system of a true *Pereskia* species.

### 3.3. Pollinators, Pollinator Behavior, and Pollinator Frequency

Overall, the flowers of *C. hildmannianus* show a clear sphingophilous syndrome; with nocturnal anthesis, long floral tube length, high nectar volume, and around 20% of sugar concentration [24]. In agreement with the literature, only hawk moths (Sphingidae) were recorded as the pollinators of *C. hildmannianus*. This confirms our hypothesis for this species’ pollination and is also in line with precedent literature for the genus [19,20,25]. Sphingidae hawk moths were recorded as the only pollinators of *C. peruvianus* [19], *C. fernambucensis* [20], and *C. aethiops* [25]. All these three *Cereus* spp. share the same set of flower features: nocturnal, nectariferous, tubulose, fragrant flowers with a perianth color varying from white to light pink. Pollinator behavior was also described similarly; with the Sphingidae uncoiling the proboscis still in flight and touching the stigma and stamens with the ventral part of the body. These preceding reports, however, didn’t give further details on insect permanency inside the floral tube [19,20,25]. In this regard, this study provides a more complete record of pollinator behavior, permanency, and frequency at the flowers. In contrast, Nasser et al. (1997) reported bat pollination in the Caribbean *Cereus repandus* Haw. but without giving details on pollinator behavior. Based on the prevalence of the observed flower features, we expect that all species of *Cereus* and *Harrisia* (Trichocereae) are pollinated by Sphingidae moths. One of us (OPB), has recently filmed unidentified Sphingidae pollinating flowers of *Cereus hexagonus* (L.) Mill. in the Colombian Andes. At least two other cactus species from Rio Grande do Sul present sphingophilous flower features: *Epiphyllum phyllanthus* (L.) Haw. and *Echinopsis oxygona* (Link.) Zucc. Globally, 24% of the published pollination reports in Cactaceae involve moths [21].

The pollination strategy of *P. aculeata* reasonably matches with mellitophily, since the flowers present diurnal anthesis, short floral tube, low nectar volume, high pollen production, and a pleasant smell [24]. Whereas native bees were the main pollinators of *P. aculeata*, insects of other orders were also recorded as pollinators (albeit less important in terms of frequency). Hence, we prefer to consider the flowers of *P. aculeata* as more generalistic in terms of pollination. It is important to stress that among the ten recorded pollinators, eight are bees and seven of these are native. Solitary native bees have already been documented as the main pollinators of sympatric yet unrelated Cactaceae [7,8,9]. Globally [21], bees are the most important group of cactus pollinators, accounting for ca. 83% of cactus pollination reports. Remarkably, the most frequent pollinators we observed were native, social Meliponini bees; these were followed by (introduced) Honeybees, two large Bumblebee species, and a large carpenter bee, respectively (Table 2). As a whole, these bees markedly differed regarding their behavior towards flowers. All recorded bee pollinators were seen walking or landing on the stigmatic surface of the same flower where they were foraging. Since *P. aculeata* is self-compatible, this behavior clearly promotes some pollinator-mediated self-pollinations. The longer the insects stay at the flowers, the higher the chances of promoting self-pollination [34]. In this regard, Meliponini bees accounted for the longer, more passive visits, together with *Caryedes* beetles (Table 2), and both insect groups may promote a considerable degree of self-pollination. On the other hand, large bees tended to stay for considerably shorter times at the flowers (Table 2). Since bumblebees and carpenter bees have a large corporal surface to adhere the pollen and stay for brief times at the flowers, we consider them not only effective pollinators but the most likely to promote cross-pollination. *Bombus morio* can indeed visit twice the flowers as *A. mellifera* per minute [35].

Non-Apidae insects had a lower number of interactions compared with native bees and *A. mellifera*. The beetle *Caryedes* sp. forages on pollen. Their behavior hardly promotes cross-pollination because individuals spend a long time in the same flower, usually using tepals as a place for reproductive interactions. Due to the long duration of floral visits, these beetles may mostly promote self-pollination. On the other hand, two Lepidoptera species, *Urbanus* sp. (Hesperiidae) and *Aellopus* sp. (Sphingidae), were eventually recorded feeding on nectar. Only one visit was observed for each species. Thus, whereas they cannot be excluded as pollinators, their rareness makes it improbable that they consistently contribute to the fruit set of *P. aculeata*.

## 4. Materials and Methods

### 4.1. Study Species

*Cereus hildmannianus* L. is a columnar (up to 15 m high) cactus (Figure 1a). This species is widespread in Southern and Southeastern Brazil [3]. The flowers are long (up to 20 cm in length) and their coloration varies from white to clear pink [3] (Figure 1a–c). The fruits of this species are smooth, spineless, and edible [10]. In contrast, *P. aculeata* Mill. is a sprawling climber, with well-developed ovate leaves (Figure 1d,e). This species is also well-distributed in Southern and Southeastern Brazil; dwelling also in Northern Argentina and several other countries in tropical America [3,22]. Flowers are dish-like, ca. 5 cm in diameter. The perianth varies from white to light cream in color (Figure 1d–f). The stamen filaments vary from light yellow to red [3] (Figure 1d–f). Unlike most Cactaceae, the flowers of *P. aculeata* present a superior ovary (not inferior, the predominant condition in Cactaceae) [22].

### 4.2. Study Sites

The studies were conducted in three localities. All areas are natural occurrences for the respective species [3] and are located in the State of Rio Grande do Sul, Southern Brazil. *Cereus hildmannianus* observations were conducted in the rural area of Caçapava do Sul municipality (30°39′04.9″ S, 53°34′34.9″ W) and São Francisco de Assis (29°33′16″ S, 55°07′38″ W). *Pereskia aculeata* was studied at the Porto Alegre Botanical Garden within the Porto Alegre municipality (30°03′13.9″ S, 51°10′39.4″ W). While Caçapava do Sul and São Francisco de Assis are placed within the Pampa Biome, Porto Alegre is located in the transitional area between the Pampas and the Atlantic Rain Forest.

### 4.3. Nectar Concentration

Nectar features were measured in situ from five flowers randomly chosen for each individual (four in *P. aculeata* and ten in *C. hildmannianus*) and bagged before anthesis. At the beginning of the anthesis, we measured the volume and sugar concentration (total sugars) of nectar with a syringe and a manual refractometer.

### 4.4. Reproductive Biology

Breeding system experiments were performed in situ and we used voile bags to hide the flowers before anthesis to isolate them from visitors and pollinators. The same treatments were applied to both study species. (1) Manual cross-pollination (flowers were pollinated with pollen from another individual in at least 100 m of distance); (2) manual self-pollination (flowers were pollinated with their own pollen); (3) apomixis (anthers were cut off); and (4) spontaneous self-pollination (intact flowers were bagged without any further manipulation). The number of individuals and flowers per treatment used in the experiments is summarized in Table 1. Since *C. hildmannianus* flowers gradually produce a limited number of flowers per night, the flowers of 16 individuals were used in experimental treatments (Table 1). All experimental treatments in *P. aculeata* were applied to four individuals (Table 1). After manipulation, flowers were bagged and the fructification rate was documented when fruits ripened. No statistical comparisons were necessary for the results concerning *C. hildmannianus* (see Section 2). Treatments yielding fruit in *P. aculeata* were statistically compared by using the non-parametric Kruskal–Wallis test.

Natural pollination (fruiting success of flowers exposed to pollinators) in *C. hildmannianus* and *P. aculeata* was measured by counting the number of fruits developed from 30 previously tagged flowers produced by five and four different individuals, respectively (Table 1).

### 4.5. Pollination Observations

Observations were performed during the flowering periods of both species in 2018 and 2019. Plant phenologies do not fully overlap. Anthesis also happens during different hour intervals (see Results); thus, pollination biology of *C. hildmannianus* was studied during night hours (9:00 PM to 7:00 AM) and *P. aculeata* was studied during the day (6:00 AM to 1:00 PM). The pollination of *C. hildmannianus* in Caçapava do Sul was studied between January and February 2019, totaling 32 h of observation. The same species was studied in São Francisco de Assis in December 2019, totaling 25 h of observation. The natural pollination of *P. aculeata* was studied at the Porto Alegre Botanical Garden. Observations were made in March 2018 and March 2019, totaling 75 h of observation. It is important to stress that flower lifespan in both species is less than one day and that—unlike other cacti—the flowers do not open again the next day. The behavior of pollinators was documented through photos and videos. These files were used to analyze insect behavior and to determine if these animals were effective pollinators or non-pollinating visitors. For the purposes of this contribution, “pollinator” is defined as any animal pollen vector that, during foraging activities at the flowers, contacts both fertile flower whorls, thus promoting pollen transfer to the stigmas. Animals not fulfilling these requirements are thereafter considered non-pollinating flower visitors. Insects were collected for subsequent identification by specialists and vouchers were deposited at the entomological collection of the Fundação Estadual de Pesquisas Agropecuárias do Rio Grande do Sul (FEPAGRO). To ascertain pollinator frequency, in each individual of *P. aculeata* we selected five flowers to compute the duration and frequency of interactions during the anthesis. In *C. hildmannianus*, we selected one individual per night and observed all interactions. This operational difference between species is due to specific flowering characteristics of each species. As mentioned above, *P. aculeata* blooms abundantly during each blooming event whereas *C. hildmannianus* produces few flowers per night.

## 5. Conclusions

The two studied species are clearly pollinator-dependent, yet presented contrasting breeding systems (self-compatibility and self-incompatibility) and pollinators (Sphingidae and mainly melitophilous). At the beginning of this contribution, we explicitly formulated the following hypotheses: (1) That both study species should be pollinator-dependent (this is, may need the agency of pollinators to set fruit and viable seed); (2) that, owing to the rarity of fruits, *C. hildmannianus* should be self-incompatible (unable to set fruit when pollinated with pollen of the same individual); (3) that, owing to the commonness of fruits in *P. aculeata*, this species may be self-compatible; (4) that owing to its flower features and according to preceding literature in related species, *Cereus hildmannianus* may be pollinated by Sphingidae moths; and (5) that owing to its flower features, *P. aculeata* may be pollinated by bees. Owing to our results, we can conclude that hypotheses 1, 2, 3, and 4 are fully confirmed. Hypothesis 5, however, is only partially confirmed. Native bees were indeed the main pollinators during our observations, but we recorded that *P. aculeata* also benefited from wasp, Diptera, and Coleoptera pollinators. Irrespective of these results, this contribution highlights the importance of native pollinators for the reproduction of cacti native to Southern Brazil, which is in line with the few preceding reports [7,8,9]. Thus, and as already suggested for other plant groups with economic relevance [6,36], measurements are to be taken to avoid habitat loss not only to conserve plant populations but also to grant places for their pollinators to live and reproduce.

## Figures and Tables

**Figure 1 plants-12-01298-f001:**
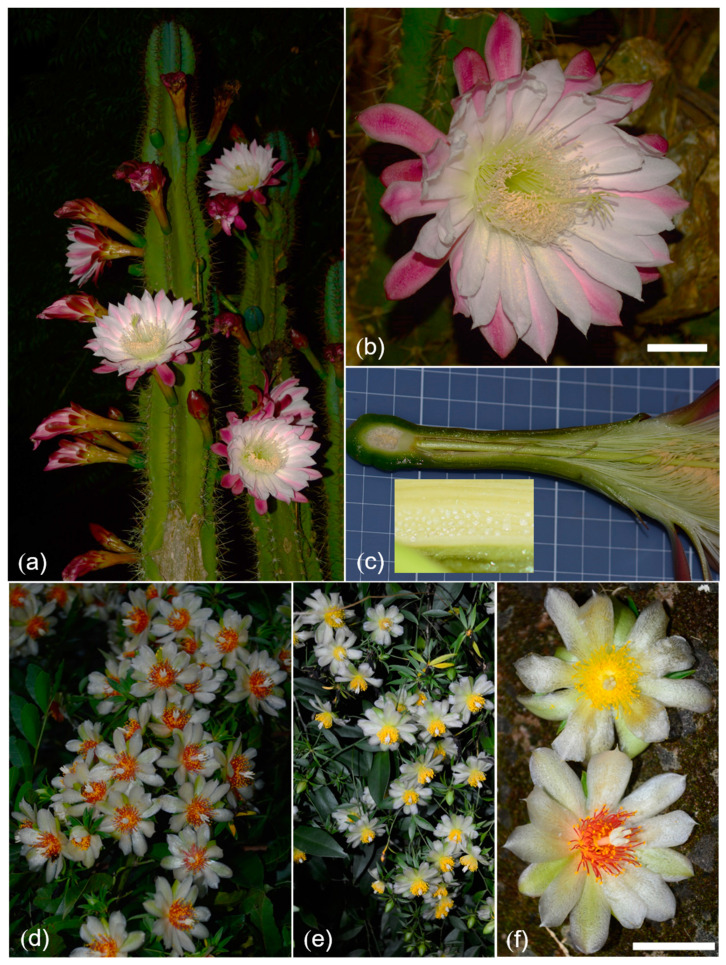
Flower features in the studied species. (**a**–**c**) *Cereus hildmannianus*. (**a**) Plant habit. (**b**) Flower in frontal view. (**c**) Flower dissected in longitudinal view. Inset: nectar droplets exuded on the hypanthium. (**d**–**f**) *Pereskia aculeata*. (**d**–**f**) Plant habit. (**f**) Frontal view of flowers with yellow and red stamen filaments. Scale bar in (**b**,**f**): 2 cm.

**Figure 2 plants-12-01298-f002:**
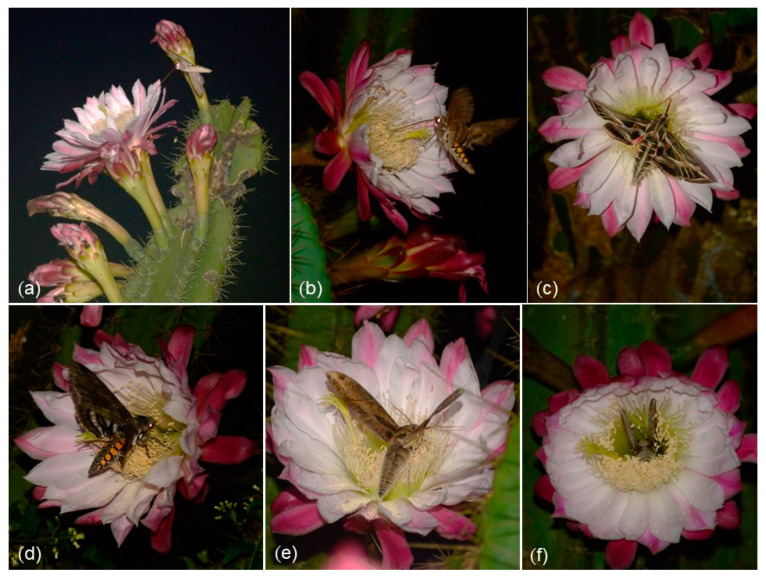
Hawk moths (Sphingidae, Lepidoptera) pollinating *Cereus hildmannianus*. (**a**,**b**) Hawk moths arriving with uncoiled proboscis. (**c**–**f**) Hawk moths entering the floral tube. (**c**) *Eumorpha vitis*. (**d**) *Manduca sexta*. (**e**) *Eumorpha satellitia*. (**f**) *E. satellitia* inside the floral tube.

**Figure 3 plants-12-01298-f003:**
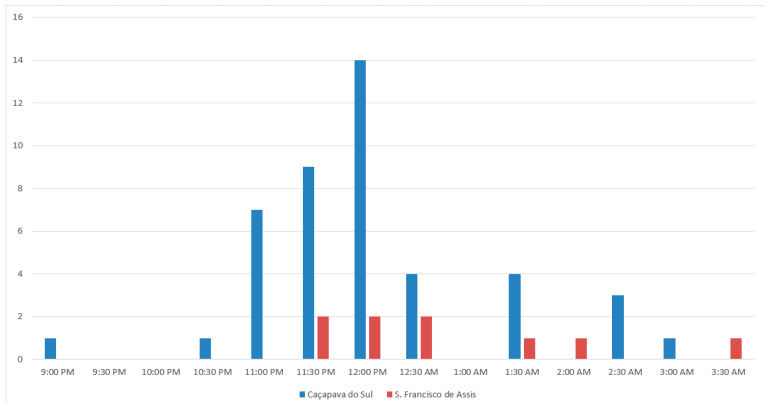
Number of hawk moths interactions in half-hour intervals during anthesis of *Cereus hildmannianus* in two different localities: Caçapava do Sul (five nights) and São Francisco de Assis (four nights), Rio Grande do Sul State, Southern Brazil.

**Figure 4 plants-12-01298-f004:**
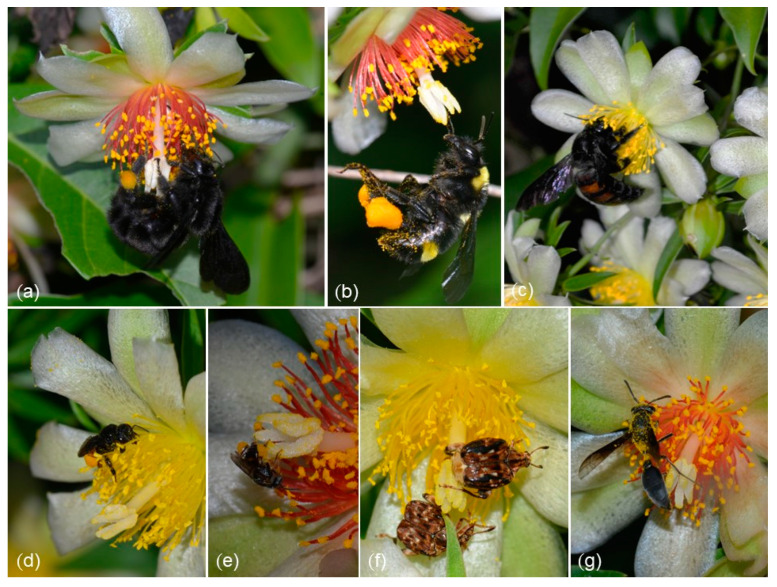
Diversity of pollinators in *Pereskia aculeata*. (**a**) *Bombus morio*. (**b**) *Bombus pauloensis*. (**c**) *Xylocopa frontalis*. (**d**) *Scaptotrigona bipunctata*. (**e**) *Plebeia droryana*. (**f**) *Caryedes* sp. (**g**) *Polybia* sp.

**Figure 5 plants-12-01298-f005:**
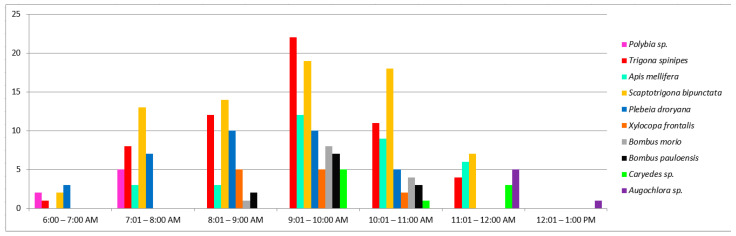
Number of species interactions in one-hour intervals during anthesis of *Pereskia aculeata* in Porto Alegre Botanical Garden (10 mornings).

**Table 1 plants-12-01298-t001:** Breeding system treatments in *Cereus hildmannianus* and *Pereskia aculeata*. Both species are pollinator-dependent since no fruits were obtained either from intact or emasculated flowers. *Cereus hildmannianus* is self-incompatible and *Pereskia aculeata* is fully self-compatible (*n* = number of individuals; numbers in parentheses represent the number of fruits obtained on the number of flowers used in each treatment).

Treatment	*Cereus hildmannianus* (*n* = 16)	*Pereskia aculeata* (*n* = 4)
Control	0% (0/30)	0% (0/30)
Emasculation	0% (0/30)	0% (0/30)
Self-pollination	0% (0/30)	78% (24/30)
Cross-pollination	83.3% (25/30)	86.25% (69/80)
Natural pollination	70% (21/30)	100% (30/30)

**Table 2 plants-12-01298-t002:** Frequency of pollinators in *Pereskia aculeata*. V = number of visits. S = average time of visits (in seconds) and standard deviation. P = pollen. N = nectar.

Species	Foraged Resource	V	S
*Scaptotrigona bipunctata* (Apidae; Meliponini)	P, N	73	35 ± 21.1
*Trigona spinipes* (Apidae; Meliponini)	P, N	58	32 ± 17.1
*Plebeia droryana* (Apidae; Meliponini)	P, N	35	9 ± 3.4
*Apis mellifera* (Apidae; Apini)	P, N	33	7 ± 3.2
*Bombus pauloensis* (Apidae; Bombini)	P, N	13	4 ± 2.7
*Bombus morio* (Apidae; Bombini)	P, N	13	5 ± 2.4
*Xylocopa frontalis* (Apidae; Xylocopini)	P, N	12	8 ± 4.9
*Caryedes* sp. (Chrysomelidae; Bruchinae)	P	9	104 ± 28.5
*Polybia* sp. (Vespidae; Epiponini)	P, N	7	44 ± 15.6
*Augochlora* sp. (Apidae; Halictinae)	P, N	6	57 ± 29.5

## Data Availability

Data available on request from the corresponding author.

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
