# Peer review of "Contrasting Pollination Strategies and Breeding Systems in Two Native Useful Cacti from Southern Brazil"

_plants, 2023, doi:10.3390/plants12061298_

Round 1

Reviewer 1 Report

This is a useful and carefully conducted study of the basic biology of two economically important cacti from Brazil. Although the hypotheses are not innovative and the findings are not surprising, the data will be useful for conservation and management of the species, and would be of value for future studies of the biology and ecology of the species. 

Specific comments for the authors:

L123 – Table 1 – The number of individuals used for the P. aculeata experiments is very low (N=4). Although the authors note that individual plants do not produce many flowers per night, they do not say that the plants are rare, or provide any other explanation for only using four individuals in the experiments. The authors need to explain/justify this approach.

L129-148 – the video/photographic data to distinguish flower visitors from pollinators is extremely valuable, and often left out of pollination studies. Kudos to the authors.

L349-351 – given that the study areas vary in their ecological contexts: pampa vs pampa/atlantic rainforest boundary, it would be valuable to know if the pollinator / flower visitor species differed between ecological contexts.  These data could be included in Table 1.

L361 – the authors don’t specify how the pollen donors were chosen for the hand pollination treatments. As there’s evidence of self-incompatibility in C. hildmannianus, finding an unrelated pollen donor would be crucial. Apart from genotyping individuals, choosing individuals located at some distance from the pollen recipient rather than immediate neighbors of the pollen recipient would be a sensible strategy to avoid sampling from family groups. The authors may have done this, but they should make it clear in the manuscript.  

Author Response

Dear Reviewer,

Thank you for your valuable suggestions which made the work more concise and readable. We also appreciate your compliments about the videos. It is often not easy to get good records of pollinators, and we are happy when it is possible to publish good images and this is recognized.

We sent you a PDF file describing each specific justification for each of your suggestions.

Yours sincerely,

Rafael Becker

Reviewer 2 Report

This is an interesting set of findings, and a badly needed addition to the literature. A very nice piece of work. I suggest that you point out very early that reproductive biology of Pereskia is unstudied, a glaring omission given its basal position to Cactaceae. 

An editorial issue that should be addressed is as follows:  Human uses of these two species are discussed at length in the introduction, and this topic is not revisited in the discussion. I suggest that you shorten this in the introduction and restructure the introduction to match those topics that you stress in the discussion (which seems to be pollination syndromes). 

Overall, the phrasing is very rough and could use a thorough review by a fluent English speaker. I am not taking that job on as a reviewer and will leave it to the editor to decide on how much wordsmithing is needed before publication.

In the abstract, scientific names are not italicized/ underlined. Throughout the paper, the genus name is spelled out when it could be represented by an abbreviation after the first time it is used.

Again, I hope to see this is print soon!

Here are more detailed comments.

L17. Sphingidae hawkmoths???

L35, 37, this “ diversity of habitats ultimately reflects in the diversity of forms, sizes, and floral morphology’ could be simplified to say the same thing “diversity of habitats and of forms, sizes and floral morphology

L43 because IT holds

L46-57. Needs to be rewritten for clarity.

L77. It is not clear to the reader what “According to the preceding bibliography, ….” is referring to.

L79. Consider moving the statement “other species in the genus” to the start of this same sentence and reframing the sentence?

L80. Should the sentence ending her have a (pers. observ.)?

L82. There is a large literature that argues about assuming the pollination syndrome by the shape and color of the flower. Consider different wording or deleting this?

L83-89. Given there is no data or very little data to to support or refute these statements, can you back off here and say in this study data will be collected on aspects of the reproductive biology?

L344, Does “As a whole” mean all or most 

L364. Wording seems wrong “number of used individuals and flowers….”

L373 is a wording missing before “by countitng”?

L378. Meaning of “day periods” is not clear

L398. Word missing here

L133. Why say eventually here?

L147 “at 35-50 min intervals” suggests you had moths individually marked. Is this the case? If not, what do you mean here?

In the caption for Fig. 3. It would be good to indicate how many days or observations are represented. Also, were all those time periods sampled evenly?

L178. Word missing

L 184. Meliponini shown be in parentheses

L193-194, and l198. Please rewrite this sentence for clarity.

In the caption for Fig. 5. It would be good to indicate how many days or observations are represented. Also, were all those time periods sampled evenly?

Consider mentioning this in the abstract and introduction?“this is also 205 the first pollination report for the whole cactus subfamily Pereskioideae, “

I disagree with this statement “Generalist morphology is expected for an early diver- 217 gent group, such as Pereskia [22]. So, more specialized pollination strategies (such as orni- 218 thophily or sphingophily) are to be expected in more internal, and more recently-diverged 219 groups [23].”, given there are many derived groups with flowers that attract diverse suites 0f pollinators.  It sounds like you are implying that complexity means advanced, and we are all taught that evolution does not work this way 

L262. Should “to” be changed to “towards”

L283 Share should not be capitalized

L301. Capitalize orders

L314. Need a reference or references for sentence ending on this line

L401-4113. I would consider these finding of your research instead of potential hypotheses.

Check all references for consistency.

Author Response

Dear Reviewer,

Thanks for all your valuable suggestions which made the work more concise and readable. Your idea of rewriting some paragraphs really made a lot of sense and the text became much more practical. About your suggestion to highlight in the abstract and introduction about being the first record of pollinators in Pereskioideae, we chose not to do it. The genus Pereskia was misused in plants of the Leuebergioideae lineage and to avoid misunderstandings, we prefer to just discuss it in the Discussion.

We sent you a PDF file describing each specific justification for each of your suggestions.

Yours sincerely,

Rafael Becker
